# The Way to Malignant Transformation: Can Epigenetic Alterations Be Used to Diagnose Early-Stage Head and Neck Cancer?

**DOI:** 10.3390/biomedicines11061717

**Published:** 2023-06-15

**Authors:** Ting-Yu Lai, Ying-Chieh Ko, Yu-Lian Chen, Su-Fang Lin

**Affiliations:** 1Institute of Bioinformatics and Structural Biology, National Tsing Hua University, Hsinchu 30013, Taiwan; s108080806@m108.nthu.edu.tw; 2National Institute of Cancer Research, National Health Research Institutes, Miaoli 35053, Taiwan; ingridko@nhri.edu.tw (Y.-C.K.); cyl@nhri.edu.tw (Y.-L.C.)

**Keywords:** oral cancer, oral potentially malignant disorders, malignant transformation, epigenome, folate, phenotypic plasticity, noninvasive biomarkers, secondary prevention

## Abstract

Identifying and treating tumors early is the key to secondary prevention in cancer control. At present, prevention of oral cancer is still challenging because the molecular drivers responsible for malignant transformation of the 11 clinically defined oral potentially malignant disorders are still unknown. In this review, we focused on studies that elucidate the epigenetic alterations demarcating malignant and nonmalignant epigenomes and prioritized findings from clinical samples. Head and neck included, the genomes of many cancer types are largely hypomethylated and accompanied by focal hypermethylation on certain specific regions. We revisited prior studies that demonstrated that sufficient uptake of folate, the primary dietary methyl donor, is associated with oral cancer reduction. As epigenetically driven phenotypic plasticity, a newly recognized hallmark of cancer, has been linked to tumor initiation, cell fate determination, and drug resistance, we discussed prior findings that might be associated with this hallmark, including gene clusters (11q13.3, 19q13.43, 20q11.2, 22q11-13) with great potential for oral cancer biomarkers, and successful examples in screening early-stage nasopharyngeal carcinoma. Although one-size-fits-all approaches have been shown to be ineffective in most cancer therapies, the rapid development of epigenome sequencing methods raises the possibility that this nonmutagenic approach may be an exception. Only time will tell.

## 1. Prevention and Control of Oral Cancer by 2030

The upcoming 19th volume of the International Agency for Research on Cancer (IARC) Handbook of Cancer Prevention is enthusiastically expected. Experts of the field critically evaluated all relevant published and unpublished studies and drafted the roadmap for the way to contain oral diseases, including oral cancer and potentially malignant disorders (OPMDs), by 2030 [1,2]. Cessation of addicted exposure to three risk factors, i.e., tobacco smoking, betel quid chewing, and alcohol consumption, will be endorsed as primary prevention measures for oral cancer and OPMDs. In Taiwan, preventive polices advocated by the government and nonprofit organizations have successfully reduced the populations of male smokers and betel quid users by 32.3% and 64.5% in 10 years, respectively (Figure 1). For secondary prevention, conventional clinical oral examination, i.e., visual inspection of the oral cavity mucosa and palpation of the external facial and neck regions, remains the most reliable screening method compared with oral cytology, light-based assays, or vital staining [3]. With well-trained healthcare workers, the sensitivity and specificity estimates of clinical oral examination ranged from 50 to 99% and 75 to 99%, respectively [4]. Indeed, screening of high-risk populations significantly reduced mortality from oral cancer, as evidenced in a randomized control trial conducted in India with a 15-year follow-up [5,6,7] and a population cohort study comprising 2,334,299 Taiwanese cigarette smokers and/or betel quid chewers [8].

However, if secondary prevention is intended to identify tumors early and treat them before progression, we expect to see a temporary increase in cancer incidence followed by a reduction in cancer death in the long run. Oddly, while we observed that cancer incidence increased rapidly (then plateaued) 5 years after the implementation of biennial oral mucosal screening in high-risk populations since 2004 [10], only limited progress was made in overall mortality and survival 18 years later [11,12]. For example, from 2007 to 2019, the improvement in 5-year relative survival rates in oral cancer was inferior to that of all types combined (5.7% vs. 12.3%) and to that of U.S. oral cancer (6.6%), where no nationwide oral mucosa screening was executed (Figure 2). Therefore, what did we miss? How to define early-stage? How can genomic and epigenomic data lend a hand in identifying and curing potentially malignant lesions before they transform?

## 2. The Malignant Transformation Potential of Each OPMD Subtype Varies

Thanks to the expert group in the 2020 March symposium organized by the WHO Collaborating Centre for Oral Cancer in the UK, a risk-based classification system is now available to distinguish eleven clinically distinctive OPMDs [14,15]. This system takes into account both the clinical and histopathological features of each subtype. In addition, through systematic review and meta-analysis of relevant studies, the malignant transformation rates of certain subtypes were compared side-by-side; leukoplakia is the most common OPMD, and proliferative verrucous leukoplakia (PVL) is the most dangerous [16] (Figure 3A). Notably, when compared with the conversion rates identified from the Taiwan nationwide oral mucosa screening program published by Chuang et al. [17] and by Chiu et al. [18], leukoplakia in Taiwan seems to be less aggressive than that in other countries (Figure 3B,C). In addition, instead of PVL, exophytic verrucous hyperplasia/oral verrucous hyperplasia, collectively designated OEVH, has the greatest potential for malignant transformation. The expert group reserved the possibility that OEVH might arise as a secondary lesion in patients with oral submucous fibrosis (OSF) [15]. If this concern is true, OSF represents the riskiest subtype in Taiwan, with an estimated annual malignant conversion rate ranging from 8.6 to 12.8 per 1000 person-years [17,18]. Further studies from other betel quid chewing-endemic countries are required to resolve this uncertainty.

Taken together, with this updated information, risk stratification within each subtype can be further individualized based on patient factors including age, sex, smoking status, betel quid chewing habit, and alcohol consumption. Patients with high-risk OPMDs should be closely monitored with regular follow-up visits and biopsies as needed. However, it is worth noting that most patients diagnosed with oral cancer do not have a clinical history of OPMD [19], and the majority of OPMD lesions do not progress to carcinoma (Figure 3). Therefore, what is the “no-return point” for an oral mucosal epithelial cell on its way to malignant transformation? In the context of epigenetic alterations, can we identify DNA methylation regions representative of each key step in the well-established histological progression of head and neck cancer, including hyperplasia (*CDKN2A* inactivation), dysplasia (*TP53* inactivation), carcinoma in situ (*CCND1* amplification), and invasive carcinoma (*PTEN* inactivation) [19]?

## 3. Global DNA Hypomethylation in the Cancer Genome

Genomic hypomethylation refers to a process in which there is a decrease in the level of DNA methylation, meaning the addition of a methyl group to the cytosine base of DNA. Hypomethylation can lead to genomic instability and abnormal gene expression, which can contribute to the development of cancer. Decades before next generation sequencing was available, Feinberg and Vogelstein published an interesting paper in that of the four of five patients with colon or lung cancer studied, substantial hypomethylation was found in several specific genomic regions of three genes (growth hormone, α-globin, γ-globin) compared with their normal counterparts. In a liver metastasis from one patient with lung cancer, the degree of hypomethylation was even greater than that of the primary tumor, indicating a progressive process during metastasis [20]. In 1996, Nawroz et al. employed PCR-based microsatellite DNA analysis to detect alterations in serum DNA from 21 head and neck cancer patients. Twelve markers indicative of microsatellite instability were used, including two trinucleotide (D14S50, DRPLA), two tetranucleotide (D21S1245, FgA), and eight markers from chromosomes 3p, 9p, and 17p. They found that 29% of patients (6 out of 21) had microsatellite alterations in serum DNA matching those in primary tumors. All six patients had advanced disease (stage III or IV), five had nodal metastases, three later developed distant metastases, and four died of disease. In contrast, another group of patients without serum microsatellite DNA alterations had stage I and II cancer and were considered to have good prognoses [21].

In the early 2010s, with the applicability of whole genome bisulfite sequencing (WGBS) in clinical samples, the notion that the cancer genome is less methylated than its normal counterpart was first confirmed at a single-base resolution [22,23,24]. Hansen et al. showed that stochastic methylation variation of the same cancer-specific differentially DNA-methylated regions, termed cDMRs, can distinguish cancer from normal tissues in colon, lung, breast, thyroid, and Wilms’ tumors. Surprisingly, close to half the genome comprising cDMRs was hypomethylated, which led to extreme gene expression variability. Genes associated with cDMRs and large blocks are involved in mitosis and matrix remodeling [22]. Later studies from the same group employed WGBS to analyze the impact of transformation on the methylomes of normal B cells, activated B cells, and Epstein–Barr virus (EBV)-immortalized B cells from the same three individuals. The results showed that EBV immortalization induced large-scale hypomethylated blocks comprising two-thirds of the genome, which was not observed in B-cell activation per se [24]. These results illuminated the mechanism and timing of altered DNA methylation in human cancer, specifically in the context of EBV.

In agreement with the abovementioned findings, Chan et al. demonstrated genome-wide DNA hypomethylation and copy number aberrations in plasma DNAs of patients from multiple cancer types, including another EBV-associated malignancy, nasopharyngeal carcinoma. Their approach is particularly attractive because high sensitivity and specificity can be achieved using low sequence depth, which is practical diagnostically [23]. As a predecessor of WGBS, high-density chip arrays have been widely used to detect genome-wide hypomethylation in clinical samples. Specifically, Timp et al. used Illumina Infinium HumanMethylation450 BeadChip arrays to analyze DNA methylation patterns of 10 breast, 28 colon, 9 lung, 38 thyroid, and 18 pancreatic cancers, as well as 5 pancreatic neuroendocrine tumors and 51 premalignant lesions. They showed that large hypomethylated blocks are a universal feature of all five tumor types analyzed, as well as premalignant lesions of the breast (*n* = 4), colon (*n* = 10), pancreas (*n* = 6), and thyroid (*n* = 31), indicating that hypomethylation occurs early in cancer development. Of note, hypervariably expressed genes were enriched within hypomethylated blocks in all tumor types [25].

For head and neck cancer, Hsiung et al. assessed global DNA methylation levels in DNA derived from whole blood of 278 patients with head and neck squamous cell carcinoma (HNSCC) and 526 controls by using a modified version of the combined bisulfite conversion and restriction enzyme analysis of the LRE1 sequence, a long interspersed nuclear element repeat region located on 22q11-q12. They found that global DNA methylation levels in whole blood are associated with increased HNSCC risk, particularly among individuals with low folate intake or high alcohol consumption [26]. The results from Hsiung et al. strongly indicated that the global DNA methylation level in whole blood may serve as a biomarker for HNSCC risk. Similarly, Smith et al. used the methylation states of LINE-1 as a surrogate of genome-wide methylation levels in tumor and mucosal samples from 119 HNSCC patients and 18 control individuals. They quantitated the LINE-1 methylation levels of each specimen by using pyrosequencing and showed that HNSCC specimens had a marked trend toward hypomethylation compared to control samples (46.8% vs. 54.0%, *p* < 0.001, Mann–Whitney). In addition, smoking but not alcohol consumption was associated with decreased LINE-1 methylation levels in both tumor and normal mucosal samples, indicating a role of smoking exposure in genome-wide DNA hypomethylation during tumorigenesis [27]. Richards et al. used the same method to determine LINE-1 element methylation levels in 26 HNSCC tumor samples and their matched normal adjacent tissues. They found a correlation between HPV negativity, increased genome instability, and loss of genome methylation in these tumor samples [28].

In addition to the transposable repetitive elements LINE-1 and Alu, Poage et al. included luminometric methylation (LUMA) as an independent indicator for global methylation states. In a total of 138 HNSCC tumors and 18 normal controls, they found that global hypomethylation in repetitive regions and hypermethylation of gene promoter regions are characteristic epigenetic alterations in cancer. Importantly, a significant negative correlation between hypermethylated CpG loci and LINE-1 methylation was revealed, suggesting that these CpGs are actively selected to maintain the tumor phenotype through DNA methylation, even as global methylation levels decrease [29].

## 4. Folate Deficiency and Epigenetic Alterations

Based on the notion that folate pool imbalance disrupts DNA global and specific gene methylation patterns, as well as increased DNA instability and strand breaks, Madeleine A. Kane proposed that folate repletion may be an effective chemoprevention for head and neck cancer [30]. Kraunz et al. investigated the association between dietary folate and methylation of the upstream regulatory region of *p16* in 169 HNSCC samples. They showed that a low folate diet may increase the risk of HNSCC by inducing epigenetic silencing of *p16* with an odds ratio (OR) of 2.3 (95% CI 1.1–4.8) when compared with those with high folate intake. In addition, this increased risk of *p16* epigenetic silencing was modified by the methylene tetrahydrofolate reductase (MTHFR) C677T allele [31].

MTHFR C677T is a functional polymorphism of MTHFR carried by approximately 10% of the population worldwide. MTHFR C677T was shown to result in activity reduction and impaired folate metabolism. By using quantitative liquid chromatography, Friso et al. assessed genomic DNA methylation of peripheral blood mononuclear cells from 105 subjects with T/T genotypes and 187 with C/C genotypes. The results showed that genomic DNA methylation directly correlates with folate status and inversely with plasma homocysteine levels, reinforcing that the MTHFR C677T polymorphism influences DNA methylation through an interaction with folate levels [32]. 

In a later study using pooled analysis of individual-level data from ten case–control studies participating in the International Head and Neck Cancer Epidemiology (INHANCE) Consortium, including 5127 cases and 13,249 controls, higher folate intake was associated with a reduced risk of cancers of the oral cavity, oropharynx, hypopharynx, and larynx (pooled OR 0.62, 95% CI 0.39–0.98). In addition, heavy drinkers with low folate intake had a significantly higher risk of oral cavity and pharyngeal cancer than those with never/light drinkers with high folate (OR 4.05, 95% CI 3.43–4.79) [33]. Interestingly, an association between the MTHFR C677T polymorphism and head and neck cancer was also noted in heavy drinkers in a meta-analysis of 23 case–control studies comprising 14,298 subjects [34]. Taken together, strong evidence has shown that the genome of many cancer types, including head and neck cancer, is globally hypo- but focally hyper-methylated at certain specific regions. How can these paradoxical epigenetic regulations be explained?

## 5. Global DNA Hypomethylation and Focal Hypermethylation Reside in the Same Regions of the Cancer Genome

Independent experiments conducted in the Laird Lab [35] and Ren Lab [36] portraited the epigenetic changes of the cancer epigenomes at single-base resolution in which global DNA hypomethylation and focal hypermethylation were shown to be concurrent events. Berman et al. employed reduced representation bisulfite sequencing, a variant of WGBS, to identify differentially methylated regions in 125 colorectal tumors and 25 normal tissues. They found that tumors exhibit a characteristic pattern of DNA methylation that involves the accumulation of both hypermethylation and hypomethylation alterations within large genomic domains known as partially methylated domains. They showed regions of focal hypermethylation primarily at CpG islands concentrated within regions of long-range hypomethylation, and the extent of hypermethylation and hypomethylation were highly correlated within each tumor, indicating the presence of a single cancer cell population that accumulates both alterations simultaneously [35].

Hon et al. combined multiple technologies, including WGBS, ChIP-seq, MethylC-seq, and RNA-seq, to investigate genome-wide DNA methylation patterns, specific histone modifications, and gene expression levels in the HER2-positive HCC1954 breast cancer cell line and primary mammary epithelial cells. Unambiguously, the results revealed a mutually exclusive relationship between DNA methylation and repressive chromatin marks (H3K9me3, H3K27me3) in HCC1954. They found that hypomethylated regions are biased toward gene-poor regions and have a significant fraction displaying allelic DNA methylation, where one allele is DNA methylated while the other allele is occupied by repressive chromatin marks. The authors proposed a “passive model” for global hypomethylation, which states that DNA replication errors lead to progressive loss of DNA methylation during cell division; in turn, hypomethylation might lead to increased binding of Polycomb group proteins that are known to promote chromatin compaction and gene silencing. Notably, in HCC1954 cells, the affected genes include tumor suppressors, e.g., the DNA repair gene *MGMT*, the deleted colorectal carcinoma gene *DCC*, and the deleted liver cancer gene *DLC1* [36].

Taken together, the aforementioned seminal studies and many findings alike [22,24,25,29,31] laid the foundation that loss of methylation in large blocks and concomitant accumulation of gene methylation in specific loci is a characteristic of most cancer types. However, how do these epigenetic changes manifest in cellular phenotypes? in a pathological tissue slide? and in a tumor admixed with stromal cell populations?

## 6. Epigenetic Alterations, Phenotypic Plasticity, and Malignant Transformation

In the 2022 sequel to the seminal work “Hallmarks of Cancer” published by Hanahan and Weinberg in 2000, “unlocking phenotypic plasticity” is recognized as a new emerging hallmark of cancer. This hallmark refers to the ability of cancer cells to escape from the state of terminal differentiation [37]. Feinberg and Levchenko defined phenotypic plasticity as “the ability of isogenic cells to adopt different phenotypic states transiently”, which is linked to epithelial-mesenchymal transition (EMT), drug resistance, and increased cell proliferation [38]. Emerging evidence from integrated genomic and pathological studies of clinical samples strongly suggests that epigenetic alterations are closely associated with phenotypic plasticity. A highly notable example in support of this connection came from single-cell RNA sequencing of 5902 malignant and nonmalignant cells from 18 treatment-naïve oral cancer specimens, including 5 matched lymph node metastases [39]. The study found that oral squamous cancer cells at the invasive margins adopt a partial epithelial-to-mesenchymal transition state (p-EMT) that lacks core EMT transcription factors (ZEB1, ZEB2, SNAIL1, SNAIL2, TWIST1, TWIST2). Instead, they expressed genes encoding p-EMT marker proteins (PDPN, TGFBI, LAMB3, LAMC2), which were not detectable in the central core of the tumor.

Further in vitro functional assays using one of five tested HNSCC cell lines (SCC9) revealed that these cultures contain dynamic mixtures of both p-EMT^hi^ and p-EMT^lo^ cells. When p-EMT^hi/lo^ cells were FACS-purified and cultured, both reverted to mixed populations of p-EMT^hi^ and p-EMT^lo^ cells within four days, suggesting that the p-EMT program is metastable and responsive to environmental cues in vitro and in vivo [39]. Similarly, using transcriptomic data as a surrogate for epigenetic changes, we identified one p-EMT (TW2.6) and one mesenchymal (OC3) type from five Taiwanese oral cancer cell lines. In our experiments, p-EMT but not mesenchymal cells exhibited multicellular cohesiveness and spheroid invasion in vitro, as well as collective angiolymphatic and perineural invasions in an immunodeficient mouse model [40]. Integrated transcriptomic analysis of xenograft tissues revealed that p-EMT and mesenchymal cells coexisted with different subtypes of host cancer-associated fibroblasts (CAFs): inflammatory and myofibroblastic CAFs, respectively [41,42]. Thus, our data support the hypothesis that tumor and stromal cells might influence each other through epigenetically driven cellular plasticity and coevolve to form an ecosystem [37]. Modern technologies, such as single-cell epigenomic sequencing, are essential to validate this possibility.

Another important aspect to note is that during TGFβ-triggered EMT of mouse hepatocytes (AML12), there is a global increase in repressive histone marks (H3K9Me2) and increases in euchromatin and transcriptional marks (H3K36Me3, H3K4Me3) rather than alterations in DNA methylation patterns [43]. Therefore, both DNA methylation and histone modification should be considered in future epigenetic studies.

Taken together, emerging evidence indicates that epigenetic dysregulation, comprising both hypo- and hypermethylation, might play a crucial step in cancer initiation. How are these findings translated into clinical implementation? 

## 7. Noninvasive Epigenetic Biomarkers for Early Detection of Head and Neck Cancer

Since early-stage tumors are usually invisible, an ideal risk-assessment biomarker would be expected to exist in body fluids, e.g., saliva or blood, rather than in tissues. With this in mind, we first enumerated specific genomic loci, such as 11q13.3, 20q11.21, 22q11.2 for DNA gain, and 19q13.43 for DNA hypermethylation, which have been recurrently documented in clinical head and neck specimens (Figure 4). We then proceed to present successful examples in nasopharyngeal carcinoma (NPC), where cell-free plasma DNA or NP brushing were utilized as sample sources.

### 7.1. Epigenetic Alterations Documented in Clinical Samples of Head and Neck Cancer

#### 7.1.1. Amplification of 11q13.3

Gain of 11q13.3, including amplification of *CCND1* and *FADD*, was previously shown to be associated with poor clinical outcome in oral squamous cell carcinoma [44,45], in head and neck cancer patient-derived xenograft tissues [46], and in a recent proteogenomic study of 108 HPV-negative head and neck cancer tumors, of which 66 had matched normal adjacent tissues [47]. Amplification of 11q13.3 was the strongest focal somatic copy number alteration (SCNA) and was mutually exclusive with *FAT1* truncating mutations, indicating that dysregulated actin dynamics were a common functional consequence in this cohort [47]. Notably, among the nine genes in the focal region (Figure 4A), increased levels of *CCND1* are a signature marker of carcinoma in situ [19].

#### 7.1.2. Hypermethylation of 19q13.43

Hypermethylation of 19q13.43, an evolutionarily conserved imprinted gene cluster [48], has been repeatedly documented in cancer development. Lleras et al. identified a total of 958 differentially methylated CpG loci across the human genome in primary tumor and matched mucosal samples of 45 patients with oropharyngeal squamous cell carcinoma. Of particular interest were the high-magnitude differentially methylated CpG loci identified on chromosome 19, which included a cluster of Krüppel-type zinc finger protein (ZNF) genes. ZNF genes are known to exist in the human genome as transcriptional regulators, but many of their downstream targets remain unknown. The hypermethylation of these ZNF genes on chromosome 19q13, including *ZNF542*, *ZNF667*, *ZNF471*, *ZNF671*, *ZNF447*, *ZNF132*, (Figure 4B) may play a role in the development and progression of oropharyngeal squamous cell carcinoma [49]. Of note, copy number loss of 19q13.43 was also detected in head and neck cancer-derived xenograft tissues (Figure S3 in ref [46]).

In a hospital-based cohort study, Cheng and colleagues enrolled 171 adult patients who had normal oral mucosa or OPMDs between 2012 and 2014 and followed up over time to monitor disease progression until 2017. Hypermethylation levels of *ZNF582* were measured in DNA extracted from oral swabs from these patients. After adjustment for all factors using the Cox proportional hazards model, the adjusted hazard ratio of malignant transformation for *ZNF582* hypermethylation was 11.41 (95% CI, 2.05–63.66; *p* = 0.005), suggesting that *ZNF582* methylation is an effective and noninvasive biomarker for identifying oral lesions with a high potential for malignant transformation [50]. An independent retrospective study comprising 132 patients with OPMD or oral cancer and 69 control subjects concluded that the performance of the *ZNF582* methylation assay was equivalent to that of an experienced oral maxillofacial surgeon and thus can serve as a useful diagnostic to increase the efficacy in identifying oral malignant lesions [51].

#### 7.1.3. Amplification of 20q11.21

Increased levels of DNMT3B, a de novo DNA methyltransferase, were a recurrent event detected in oral cancer tissues [52,53,54,55] as well as oral premalignant lesions [56,57], strongly supporting its role in tumorigenesis. In the vicinity of *DNMT3B*, an amplification hotspot comprising two stemness-related genes, *ID1* and *BCL2L1* (Figure 4C), is commonly found in human embryonic stem cell cultures. ID1 is a conserved negative regulator that suppresses differentiation and sustains embryonic stem cell self-renewal; BCL2L1 (a.k.a. BCL2-XL) is an antiapoptotic protein that provides a strong selective advantage for stem cell survival [58,59]. Importantly, hypomethylation of the DNMT3B promoter was noted in a recent epigenomic/transcriptomic analysis of gingivo-buccal oral squamous cell carcinoma [57]. Therefore, it is tempting to envisage that concurrent increased expression of *ID1*, *BCL2L1*, and *DNMT3B* might be a crucial step for cell fate determination during tumorigenesis. Again, amplification of this region was also detected in head and neck cancer-derived xenograft tissues (Figure S3 in ref [46]).

#### 7.1.4. Amplification of 22q11.2

Copy number gains of the potential oncogene *CRKL*, located at 22q11.2 (Figure 4D), were observed in laryngeal carcinoma [60] and 180 paired head and neck cancer samples from two high incidence regions of Europe and South America, which underwent targeted sequencing [61]. Amplification of this region was associated with decreased overall survival. It is worth noting that epigenetic alterations in locus 22q13.1, specifically hypermethylation of *H1F0* and increased expression of APOBEC3A in APOBEC deletion oral cancer carriers, were linked to functional intratumor heterogeneity [62] and patient overall survival [63], respectively.

### 7.2. Noninvasive Epigenetic Biomarkers for NPC Screening

Epstein–Barr virus (EBV) is a human oncogenic herpesvirus associated with a variety of nonmalignant, premalignant, and malignant diseases (e.g., Burkitt lymphoma and NPC). Approximately the same time that large-scale hypomethylated blocks coupled with small-scale hypermethylated CpG islands were demonstrated in EBV immortalized B cells [24], genome-wide hypomethylation and copy number aberrations were demonstrated in plasma DNA of patients with NPC by using shotgun massively parallel bisulfite sequencing. This method achieved a sensitivity and specificity of 74% and 94%, respectively, for the detection of nonmetastatic NPC patients [23]. Nevertheless, genome-wide bisulfite sequencing and technologies alike are costly and require substantial computational power. Researchers in the field have found alternative methodologies to fulfill the same goal based on understanding the EBV etiology in nasopharyngeal cancer.

During the course of establishing a representative NPC cell system for EBV infection, Li et al. found that in telomerase-immortalized nasopharyngeal epithelial cells, there was a marked increase in the expression of *p16* at the second senescent stage (37th population doubling). A complete loss of *p16* expression was detected after emergence from the second senescence stage. Significantly reduced RASSF1A and increased ID1 protein levels were observed after the loss of *p16* expression, suggesting that these are later events in the immortalization process. Cytogenetic analyses confirmed that these immortalized cells are largely diploid except homologous deletion of *p16* (9p21.3) and a clonal gain of 17q21-25 on chromosome 11p15. Together, the data indicated the presence of certain genomic alterations in premalignant nasopharyngeal tissue prior to EBV infection [64]. A later study conducted by Birdwell et al. further identified over 13,000 differentially methylated CpG residues in telomerase-immortalized oral keratinocytes exposed to EBV compared to uninfected controls. Interestingly, genes with increased transcript levels frequently acquired DNA methylation within the gene body, while those with decreased transcript levels acquired DNA methylation near the transcription start site. [65].

EBV resides in two forms during its life cycle, latent and lytic. Fernandez et al. provided evidence that EBV-encoded LMP1 and EBNA3C, previously known to increase the expression levels and stability of host DNMTs, might participate in progressive methylation of the EBV genome during latent infection. In 22 EBV DNA methylomes obtained from a collection of lymphoid samples that included both benign and malignant lymphoid disorders, primary tissues and cell lines of EBV-associated lymphomas, and EBV-positive NPC tumors, they observed that the EBV genome present in cells corresponding to benign diseases (e.g., reactive lymphadenitis in tonsils and infectious mononucleosis) showed the earliest presence of methylated EBV transcription start sites. Remarkably, the methylation of the EBV genome was significantly increased in nasopharyngeal samples of primary tumors, indicating that the latent EBV genome is a target of host methylation machinery [66]. Interestingly, as a countermeasure, others and our lab found that the immediate-lytic protein Rta interfered with the binding sites of CTCF, a host chromatin modulator on both host and viral genomes during productive replication in cells of epithelial origin [67,68].

To identify appropriate NPC epigenetic biomarkers for clinical use, a group of scientists at the University of Hong Kong performed methylation-sensitive resolution melting (MS-HRM) assays to quantitate the methylation states of four hypothetical tumor suppressors, i.e., *RASSF1*, *WIF1*, *DAPK1*, and *RARB*, in 220 and 50 plasma samples from NPC patients and control individuals, respectively. They demonstrated that this protocol can detect 63.6%, 86.7%, 88%, and 96.5% of patients with stage I, II, III, and IV tumors, respectively, with a specificity of 88%. Receiver operating characteristic (ROC) curve analysis of four genes combined had an area under the curve value (AUC) of 95.8%, indicating excellent diagnostic performance [69]. In addition, other assays alike [70] as well as plasma EBV DNA are all clinically available for NPC screening [71]. These encouraging studies have illuminated the way for biomarker development applicable to other anatomic sites of head and neck cancer. 

## 8. Future Directions

An ideal risk assessment biomarker should demonstrate high sensitivity and specificity. The success of such a biomarker is closely tied to proper specimen stratification and methodological choices during the exploratory phase. Previous studies have predominantly focused on distinguishing epigenetic alterations between tumors and adjacent normal tissues, with limited studies delving into the malignant progression of OPMD tissues. Therefore, it is imperative to establish a comprehensive and OPMD-specific epigenetic landscape. To achieve this, it is necessary to retrospectively stratify archived OPMD specimens into progressors and nonprogressors and construct epigenetic landscapes for each group so that the malignant transformation of OPMD can be captured. Moreover, the development of OPMD cell culture models, such as immortalized oral keratinocytes, combined with the transformation of these cells using IARC carcinogens such as arecoline or tobacco-specific nitrosamines, is essential. Analyzing the phenotypic and epigenetic changes in each matched cell model will provide valuable insights into identifying representative epigenetic alterations during key steps of malignant transformation.

We are currently in an exciting era of sequencing technologies with the maturation of third-generation sequencing platforms such as Oxford Nanopore Technology and Pacific Biosciences. These advanced platforms enable single-molecule, long-read sequencing that directly detects 5-methylcytosine (5mc) without the requirement of bisulfite conversion. In addition, innovative “six-base sequencing” allows simultaneous sequencing of genetic (A, G, C, T) and epigenetic (5mc, 5hmc) bases in DNA samples [72]. These cutting-edge technologies eliminate DNA fragmentation caused by bisulfite treatment, resulting in improved accuracy of DNA methylation detection and enhanced read alignment efficiency compared to whole genome bisulfite sequencing. Recently, advancements in these emerging technologies have shed light on previously undetectable epigenetic changes, such as hypermethylation in CpG-poor regions serving as a primary driver of leukemia chemotherapy resistance [73], extensive methylation heterogeneity were evident in heterochromatin regions using single-molecule analysis [74], and the landmark achievement of applying six-base sequencing in a test liquid biopsy from a patient with stage III colon cancer [72].

Once the epigenetic malignant transformation landscape of OPMD tissues is established, we will proceed to the validation phase. In this stage, cost-effective bisulfite conversion PCR-based assays will be used to validate each differentially methylated region of interest in large-scale archived samples. As our goal is to develop noninvasive biomarkers, validation assays should prioritize the use of DNAs obtained from oral swabs, saliva, or blood as sample sources. In addition, the clinical and pathological information of each sample is crucial to enhance the sensitivity and specificity of candidate biomarkers. For instance, different subtypes of OPMD exhibit varying rates of malignant transformation; therefore, the follow-up time frame should be adjusted accordingly, being shorter for more aggressive types, such as PVL, and longer for more indolent types, such as OLL (Figure 3). Considering that a significant portion of oral cancers do not have preceding OPMD lesions, it is prudent to include samples from oral cancer patients at this point. Similarly, we can incorporate previously documented epigenetically altered gene clusters from head and neck cancer tissues (Figure 4) into the mutational landscape of OPMD and distilled significant genes for validation assays. We have witnessed the success of implementing four methylated promoters for early-stage NPC diagnosis from plasma DNA samples (220 patients and 50 controls, ref [69]) and the predictive capability of *ZNF582^m^* in OPMD malignant transformation from oral swab samples (171 patients with matched OPMD and normal mucosa in ref [50]; 201 patients in ref [51]). Therefore, it is highly likely that the approaches proposed above will lead to the discovery of even more effective epigenetic biomarkers that are clinically applicable.

As a final note, the ultimate objective of secondary prevention in cancer control is to identify and treat potentially malignant lesions or early-stage tumors. A comprehensive epigenetic mutational landscape of OPMDs plays a crucial role in achieving this goal from multiple perspectives. For example, folate insufficiency and the MTHFR C677T polymorphism have been associated with genome-wide hypomethylation and an increased risk of head and neck cancer (as discussed in Section 4). Therefore, the repletion of dietary folate is an essential therapeutic intervention for patients showing indications of low folate uptake. Furthermore, promoter silencing of *H1F0* at 22q13.1 (Figure 4D) leads to a reduction in linker histone H1.0 levels, resulting in the relaxation of chromatin compaction and the upregulation of cell self-renewal genes [62]. Significantly, a clinically well-tolerated HDAC inhibitor called quisinostat has recently demonstrated the ability to effectively restore H1.0 levels in various patient-derived cancer cell lines, as well as a significant impairment of self-renewal capacity in breast and non-small lung cancer cell models [75]. This type of drug is particularly attractive for head and neck cancer, an entity of diseases prominent for field cancerization. Considering the upstream regulatory role of the epigenome in gene expression, the comprehensive epigenetic landscape of OPMD might also shed light on the interpretation of some chemopreventive drugs that were controversial in the past [76]. This includes antioxidants and compounds affecting cell fates. For instance, studies have shown that a glutathione-driven antioxidant pathway is beneficial for cancer initiation at the premalignant stage [77], and the preferential expression of antioxidant genes in keratinocytes is necessary to maintain their undifferentiated state [78]. These phenomena may be linked to epigenetically driven plasticity, and a deeper understanding of epigenetic alterations during malignant transformation is required to unravel these relationships.

## 9. Conclusions

Although there is a great variability in clinical presentations and malignant conversion rates, all eleven subtypes of oral potentially malignant disorders (OPMDs) have the potential to progress to oral cancer. Therefore, identifying predictive biomarkers for secondary prevention of oral cancer is a challenging task. Interestingly, emerging evidence suggests that biomarkers derived from epigenetic alterations, specifically hyper or hypomethylated DNAs, hold great potential for overcoming this problem. We discussed several advantages of using such examples, including that the materials can be noninvasively acquired from body fluids, DNA is more stable than other biomolecules, and most importantly, dramatic changes in the epigenomes of precancer lesions have been previously documented in many types of cancer. Using modern epigenetic sequencing technologies in clinical oral cancer and OPMD specimens should soon allow us to identify key regions of the epigenome that drive the conversion of nonmalignant oral keratinocytes into malignantly uncontrollable ones.

## Figures and Tables

**Figure 1 biomedicines-11-01717-f001:**
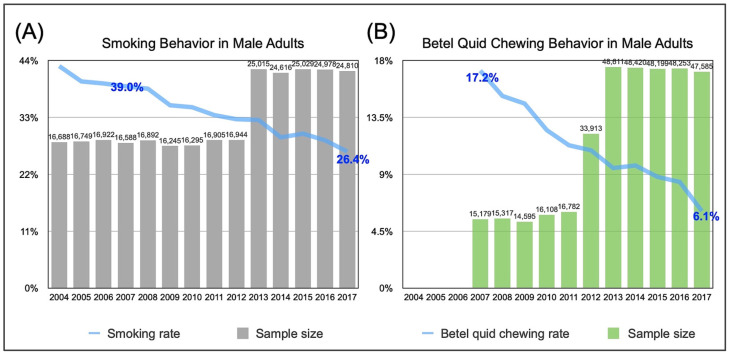
The percentages of Taiwanese male adults aged 18 or above who were smokers (**A**) or betel quid users (**B**) over the indicated years. Source of data: Taiwan Government Open Data [9].

**Figure 2 biomedicines-11-01717-f002:**
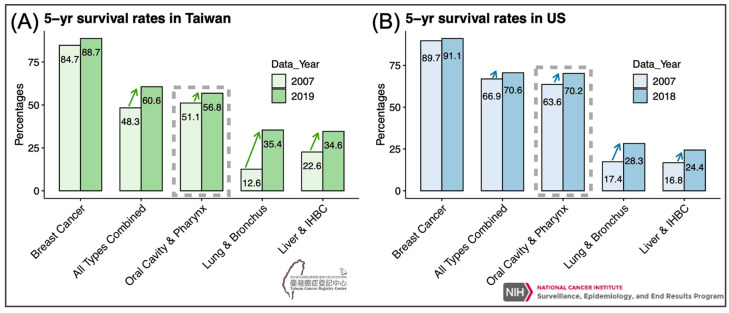
Bar plots denote the relative 5-year survival rates of the indicated cancer types in Taiwan (**A**) and in the United States (**B**). Source of data: Taiwan Cancer Registry Center [11] and SEER*Explorer [13].

**Figure 3 biomedicines-11-01717-f003:**
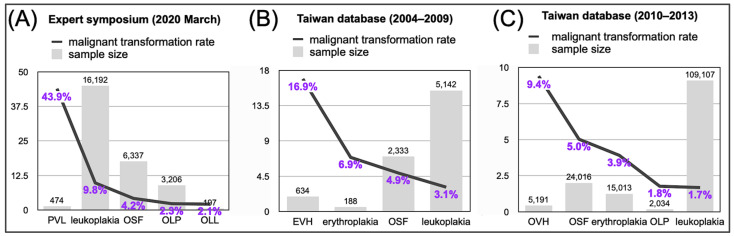
Bar charts denote malignant transformation rates of the indicated OPMD subtypes. Source of data: (**A**) Warnakulasuriya et al. 2021 [16], (**B**) Chuang et al. 2018 [17], (**C**) Chiu et al. 2021 [18]. EVH, exophytic verrucous hyperplasia; OLL, oral lichenoid lesion; OLP, oral lichen planus; OSF, oral submucous fibrosis; OVH, oral verrucous hyperplasia; PVL, proliferative verrucous leukoplakia.

**Figure 4 biomedicines-11-01717-f004:**
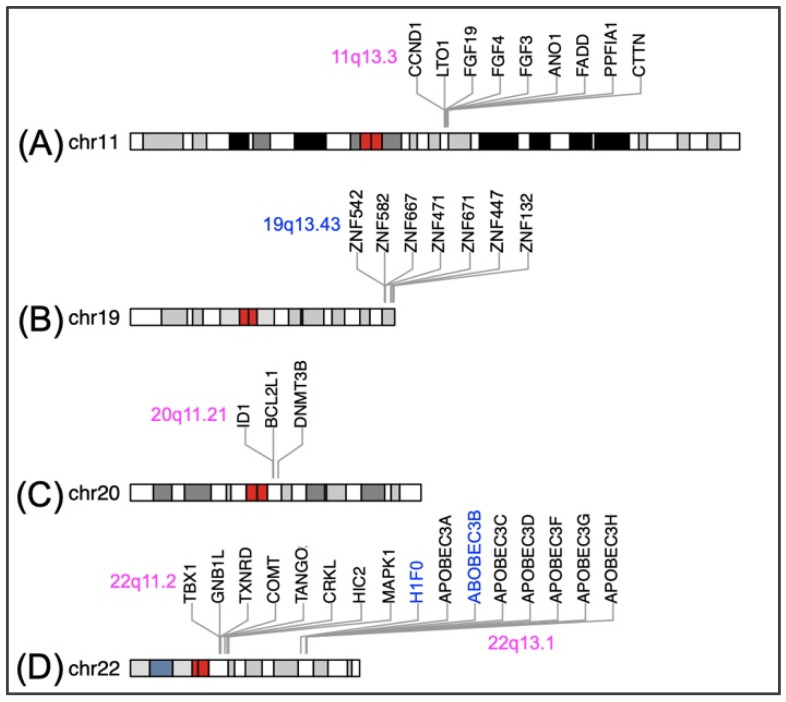
Genomic locations of the indicated genes on human chromosomes 11 (**A**), 19 (**B**), 20 (**C**), and 22 (**D**). In each ideogram, the red areas represent centromeres, while the dark regions indicate compacted chromatin.

## Data Availability

Not applicable.

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
