# Peer review of "The Way to Malignant Transformation: Can Epigenetic Alterations Be Used to Diagnose Early-Stage Head and Neck Cancer?"

_biomedicines, 2023, doi:10.3390/biomedicines11061717_

Round 1

Reviewer 1 Report

Manuscript entitled "The way to malignant transformation: focus on the epigenome"

Unfortunately, this work is superficial, rough, and out of scope. Major issues:

1. In the review, there is almost nothing related to epigenome. All discussion are not focusing and superficial.

2. The review is distractive and less updating. It is of few scientific value.

Acceptable

Reviewer 2 Report

The paper titled "The way to malignant transformation: focus on the epigenome" presents an overview of the potential identification of sensitive and highly specific oral cancer predictive biomarkers through the growing field of cancer epigenomics. While the paper provides some valuable insights, there are certain aspects that could be improved to enhance its critical analysis.

Firstly, the authors should elaborate on their proposed future trends in the field. It is essential to identify the areas where further research is needed and the potential directions for future studies. This could involve discussing emerging technologies or methodologies that could be utilized to better understand the epigenetic mechanisms underlying oral cancer development and progression. By providing clear suggestions for future trends, the paper would become more forward-thinking and provide a roadmap for researchers in the field.

Furthermore, the authors should outline a specific strategy for investigating these future trends. They need to discuss the experimental design, sample size, and selection criteria that would be required to validate the potential oral cancer predictive biomarkers identified through epigenomic analysis. It is important to emphasize the need for rigorous scientific methodologies and large-scale studies to establish the reliability and reproducibility of these biomarkers. Without a well-defined strategy, the proposed future trends may remain abstract and impractical.

In terms of implementation, the authors should address the practical challenges associated with translating the knowledge gained from epigenomic studies into clinical practice. They need to discuss the feasibility of incorporating these biomarkers into routine screening and diagnostic procedures for oral cancer. This could involve addressing issues such as cost-effectiveness, scalability, and the development of user-friendly tools or assays. By providing a detailed plan for implementation, the authors would demonstrate a realistic approach to applying their findings in a clinical setting.

Overall, the paper has the potential to make significant contributions to the field of oral cancer research. However, by expanding on future trends, proposing a clear strategy, and addressing implementation challenges, the authors can enhance the practical relevance and impact of their work.

Round 2

Reviewer 1 Report

Apologize but I don't think this work acceptable for publication. The content is superficial and not significant. 

Minor revision is suggested.